# Treatment of stimulant use disorder: A systematic review of reviews

Claire Ronsley[1,2], Seonaid Nolan [1,2], Rod Knight[1,2], Kanna Hayashi[1,3], Jano Klimas[1,4], Alex Walley[5,6], Evan Wood[1,2], Nadia Fairbairn [1,2]*

1 British Columbia Centre on Substance Use, Vancouver, BC, Canada, 2 Department of Medicine, University of British Columbia, St. Paul's Hospital, Vancouver, BC, Canada, 3 Faculty of Health Sciences, Simon Fraser University, Vancouver, BC, Canada, 4 School of Medicine, University College Dublin, Belfield, Dublin, Ireland, 5 Department of General Internal Medicine, Boston Medical Center, Boston, MA, United States of America, 6 Boston University School of Medicine, Boston, MA, United States of America

* nadia.fairbairn@bccsu.ubc.ca

**Data Availability Statement:** All relevant data are within the manuscript and its Supporting Information files.

**Funding:** This research was undertaken, in part, thanks to funding from a MSFHR/St. Paul's Foundation Scholar Award which supports Dr.

## Abstract

### Aims

Stimulant use disorder contributes to a substantial worldwide burden of disease, although evidence-based treatment options are limited. This systematic review of reviews aims to: (i) synthesize the available evidence on both psychosocial and pharmacological interventions for the treatment of stimulant use disorder; (ii) identify the most effective therapies to guide clinical practice, and (iii) highlight gaps for future study.

### Methods

A systematic database search was conducted to identify systematic reviews and meta-analyses. Eligible studies were those that followed standard systematic review methodology and assessed randomized controlled trials focused on the efficacy of interventions for stimulant use disorder. Articles were critically appraised using an assessment tool adapted from Palmeteer et al. and categorized for quality as 'core' or 'supplementary' reviews. Evidence from the included reviews were further synthesized according to pharmacological or non-pharmacological management themes.

### Results

Of 476 identified records, 29 systematic reviews examining eleven intervention modalities were included. The interventions identified include: contingency management, cognitive behavioural therapy, acupuncture, antidepressants, dopamine agonists, antipsychotics, anticonvulsants, disulfiram, opioid agonists, N-Acetylcysteine, and psychostimulants. There was sufficient evidence to support the efficacy of contingency management programs for treatment of stimulant use disorder. Psychostimulants, n-acetylcysteine, opioid agonist therapy, disulfiram and antidepressant pharmacological interventions were found to have insufficient evidence to support or discount their use. Results of this review do not support the use of all other treatment options.

Nadia Fairbairn. Evan Wood is supported by a Tier 1 Canada Research Chair in Addiction Medicine. Kanna Hayashi is supported by a CIHR New Investigator Award (MSH-141971), a Michael Smith Foundation for Health Research (MSFHR) Scholar Award, and the St. Paul's Hospital Foundation. Dr. Seonaid Nolan is supported by the MSFHR and the University of British Columbia's Steven Diamond Professorship in Addiction Care Innovation. Rod Knight is supported by a Scholar Award from MSFHR. A European Commission grant (701698) and Canadian Institutes of Health Research (671 397968, 422332) grants support Dr. Klimas.

**Competing interests:** No authors have competing interests.

## Conclusions

The results of this review supports the use of contingency management interventions for the treatment of stimulant use disorder. Although evidence to date is insufficient to support the clinical use of psychostimulants, our results demonstrate potential for future research in this area. Given the urgent need for effective pharmacological treatments for stimulant use disorder, high-quality primary research focused on the role of psychostimulant medications for the treatment of stimulant use disorder is needed.

## Introduction

Stimulant use and stimulant use disorder are associated with a range of health and social harms, including psychiatric and cardiovascular morbidity, infectious disease transmission (i.e. HIV and hepatitis C), drug associated crime, and homelessness [1–5]. The global prevalence of stimulant use has increased over the past decade, and there has been an alarming rise in the use of amphetamine-type stimulants in many jurisdictions. Recent estimates indicate there are approximately 18.1 million cocaine users worldwide, with the highest rates in North America (2.1 percent). From 2007 to 2017 there was an eightfold rise in methamphetamine seizures in East and South-East Asia, which has continued to increase[6]. North America however maintains the highest prevalence of methamphetamine use worldwide, at 2.1 percent of the population aged 15–64 [6]Australia has also seen rising rates of methamphetamine use with related-deaths having doubled between 2009 and 2015 [7].

The growing problem of stimulant use globally has emphasized a pressing need to expand access to evidence-based treatment for stimulant use disorder. Of those accessing publicly funded treatment for substance use disorder in the United States, less than one in five individuals (17.8%) are doing so for cocaine or other stimulant treatment [8]. Only a minority of patients seeking treatment for substance use disorders received evidence-based treatment, though true estimates are currently lacking. The pursuit of evidence-based interventions for treatment of stimulant use disorder has resulted in extensive investigation into a wide range of both behavioural and pharmacological therapies with mixed outcomes. While there is still no medication on the market, some psychosocial interventions have shown promising results.

There are a large number of systematic reviews and meta-analyses that have now been conducted on various treatment options for stimulant use disorder. Though several studies have assessed the efficacy for a range of interventions, there is little literature available that consolidates the current evidence. This systematic review of reviews aims to: (i) synthesize the available evidence on both psychosocial and pharmacological interventions for the treatment of stimulant use disorder; (ii) identify the most effective therapies to guide clinical practice, and (iii) highlight gaps for future study.

## Materials and methods

This 'systematic review of reviews' assembled the evidence for stimulant use disorder treatments from the systematic reviews of literature.

### Search strategy

Following the Systematic Reviews and Meta-Analyses (PRISMA) checklist [9] (see S1 Appendix), we conducted a systematic search of English language peer-reviewed literature of

*Pubmed*, *EMBASE*, and the *Cochrane Database of Systematic Reviews*, from inception through to November 26, 2019. Search terms were combined using appropriate Boolean operators and included subject heading terms or key words for three key themes and were tailored to fit each database requirement: stimulant use (i.e. "cocaine use/abuse/dependence/use disorder(s)", OR "amphetamine use/abuse/dependence/use disorder(s)", OR "methamphetamine use/abuse/ dependence/use disorder(s)") AND pharmacological or psychosocial intervention (i.e. "treatment" OR "intervention") AND systematic review study design (i.e. "systematic review" OR "meta-analysis"). A review of the citations of relevant published reviews was also performed by hand. Further details on the full electronic search strategy is included as a supplemental file (see S2 Appendix).

## Inclusion and exclusion criteria

We included all systematic reviews and meta-analyses of randomized controlled trials (RCT) involving human participants that examined pharmacological and/or psychosocial interventions for stimulant use disorder. Due to the changes in terminology from the DSM-IV to the DSM-V, all studies assessing stimulant 'abuse', 'dependence', and 'use disorder(s)' were included [10]. Studies were excluded if they evaluated substance use outcomes such that we could not extrapolate the results for stimulant use disorder. Studies were excluded if a more recent review existed that included all of the RCTs from the earlier review; if there were only some overlapping RCTs, both reviews were included. Studies were also excluded if they did not follow standard systematic review methodology as outlined by the PRISMA statement on systematic review and meta-analyses [9]. Studies were excluded if the intervention was not clearly defined. The search was limited to English language literature.

## Data extraction, analysis, and quality assessment

Titles and abstracts of retrieved reviews were screened to identify studies that met the inclusion criteria. Potentially eligible reviews were retrieved and the full text was assessed independently by two authors (CR and NF) for evaluation of eligibility criteria. We extracted the following data: study characteristics (e.g. review size, study methods, average duration of trial), participant characteristics (e.g. type of substance use, ethnicity, gender), intervention characteristics (e.g. type of psychosocial or pharmacological intervention, comparison group), and outcomes (e.g. main findings). Extracted data were summarized across reviews for: intervention type, review type, substance use type, and main outcome effects (see Table 2). For all reviews that included pooled effect size estimates, this data was included in our results.

All reviews that met inclusion criteria were critically appraised using an instrument adapted from Palmateer et al. (2010) [11] and based on recommendations for appraisal of interventional studies developed by the Health Development Agency [12] (see Table 1). Reviews were then categorized as: (i) data where the whole review is judged to be high quality; (ii) data where only a part of the review is judged to be high quality; or (iii) contextual or background material. Meta-analytics were not performed as much of the primary data was included in multiple reviews, reducing the value of analysis [13].

Evidence statements were then assigned based on the level of evidence available to support each statement. Those categorized as (i) or (ii) were deemed high quality and classified as 'core' reviews. Authors' conclusions from core reviews were used to determine the effectiveness of an intervention. Those categorized as (iii) were considered 'supplementary' reviews, and utilized as contextual material only. Two reviewers (CR, NF) independently appraised the evidence for each intervention and determined whether the evidence was 'sufficient', 'tentative', 'insufficient' or 'no evidence' to support or discount the intervention. There were two

**Table 1. Evidence statements and level of evidence needed to support each statement.** *

| Evidence statement | Level of evidence |
| --- | --- |
| Sufficient evidence from reviews to either support or discount the effectiveness of an intervention | Clear statement from one or more *core* reviews based on multiple robust studies, *or* Consistent evidence across multiple robust studies within one or more *core* reviews, in the absence of a clear and consistent statement in the review (s) |
| Tentative evidence from reviews to either support or discount the effectiveness of an intervention | A tentative statement from one or more *core* reviews based on consistent evidence from a small number of robust studies or multiple weaker studies, *or* |
| | Consistent evidence from a small number of robust studies or multiple weaker studies within one or more *core* reviews, in the absence of a clear and consistent statement in the review (s), *or* |
| | Conflicting evidence from one or more *core* reviews, with the stronger evidence weighted towards one side (either supporting or discounting effectiveness) and a plausible reason for the conflict, *or* |
| | Consistent evidence from multiple robust studies within one or more *supplementary* reviews, in the absence of a core review |
| Insufficient evidence from reviews to either support or discount the effectiveness of an intervention | A statement of insufficient evidence from a *core* review, *or* |
| | Insufficient evidence to either support or discount the effectiveness of an intervention (either because there is too little evidence or the evidence is too weak), in the absence of a clear and consistent statement of evidence from (a) *core* review(s), *or* |
| | Anything less than consistent evidence from multiple robust studies within one or more *supplementary* reviews |
| No evidence | No core or supplementary reviews of the topic identified, due possibly to a lack of primary studies |

*Palmateer et al. 2010 [11], modified from Ellis et al. 2003[14]

discrepancies which were resolved by a third independent reviewer (SN). Methods of searching and assessing primary literature, appropriateness of outcomes measured and conclusions, recognition of biases and steps taken to limit those biases, as well as the use of the PRISMA guidelines for systematic reviews were assessed.

## Results

The search generated 476 records. Screening resulted in 79 articles that met study inclusion criteria, of which 56 remained after removal of duplicates. Abstract and full-text screening resulted in a total of 29 reviews evaluating eleven interventions for stimulant use disorder. A summary of our search strategy is presented in the Preferred Reporting Items for Systematic Reviews and Meta-Analyses (PRISMA) Flow Diagram (Fig 1).

### Participants and characteristics of studies

Reviews included randomized, predominantly placebo-controlled trials. All included reviews followed standard systematic review methodology. Population characteristics included a higher proportion of males in most studies, and predominantly cocaine users, with few studies focusing on methamphetamine use or stimulant use more broadly. Pharmacological interventions were compared to placebo, no medication, or compared multiple medications.

**Table 2. Overview of included systematic reviews and statements of evidence.**

| Intervention | Review | Substance(s) | # RCTs | # Participants | Critical Assessment | Evidence Statement |
|---|---|---|---|---|---|---|
| **Contingency Management** | *De Crescenzo et al. 2018[15]* | Cocaine, amphetamine | 15 | 2024 | Core review | Sufficient evidence from reviews to support the effectiveness of contingency management for stimulant use disorder |
| | *Schumacher et al. 2007[16]* | Crack cocaine | 4 | 577 | Supplementary review | |
| ➢ CM vs. CBT[1] | *De Crescenzo et al. 2018[15]* | Cocaine, amphetamine | 4 | 395 | Core review | |
| | *Farronato et al. 2013[17]* | Cocaine | 8 | 972 | Supplementary review | |
| | *Lee et al. 2008 [18]* | Methamphetamine | 9 | 2037 | Supplementary review | |
| ➢ CM + CRA[1] | *De Crescenzo et al. 2018[15]* | Cocaine, amphetamine | 1 | 96 | Core review | |
| | *Schierenberg et al. 2012[19]* | Cocaine | 19 | 809 | Supplementary review | |
| | *Roozen et al. 2003[20]* | Cocaine | 4 | 173 | Supplementary review | |
| ➢ CM + Pharmacotherapy[1] | *Schierenberg et al. 2012[19]* | Cocaine | 19 | 809 | Supplementary review | |
| **Cognitive Behavioural Therapy** | *De Crescenzo et al. 2018[15]* | Cocaine, amphetamine | 7 | 813 | Core review | Insufficient evidence from review to either support or discount the effectiveness of CBT for stimulant use disorder |
| | *Harada et al. 2019[21]* | Amphetamine-type Stimulants | 2 | 210 | Supplementary review | |
| **Acupuncture** | *Mills et al. 2005 [22]* | Cocaine | 9 | 1747 | Supplementary review | Tentative evidence from reviews to discount the effectiveness of acupuncture for cocaine use disorder |
| | *Gates et al. 2008 [23]* | Cocaine | 7 | 1433 | Supplementary review | |
| **Antidepressants** | *Pani et al. 2011 [24]* | Cocaine | 37 | 3551 | Core review | Sufficient evidence from reviews to discount the effectiveness of antidepressants for cocaine use disorder |
| | *Chan et al. 2019 [25]* | Cocaine | 48 | * | Core review | |
| | *Torrens et al. 2005[26]* | Cocaine | 19 | 1180 | Supplementary review | |
| | *Chan et al. 2019 [27]* | Methamphetamine | 34 | * | Core review | Insufficient evidence from review to either support or discount the effectiveness of antidepressants for methamphetamine use disorder |
| **Disulfiram** | *Pani et al. 2010 [28]* | Cocaine | 7 | 492 | Core review | Insufficient evidence from review to either support or discount the effectiveness of disulfiram for cocaine use disorder |
| **Dopamine Agonists** | *Minozzi et al. 2015[29]* | Cocaine | 24 | 2147 | Core review | Sufficient evidence from reviews to discount the effectiveness of dopamine agonists for cocaine use disorder |
| | *Chan et al. 2019 [25]* | Cocaine | 48 | * | Core review | |
| **Antipsychotics** | *Indave et al. 2016 [30]* | Cocaine | 14 | 719 | Core review | Tentative evidence to discount use of antipsychotics for stimulant use disorder |
| | *Chan et al. 2019 [25]* | Cocaine | 48 | * | Core review | |
| | *Alvarez et al. 2013[31]* | Cocaine | 12 | 681 | Supplementary review | |
| | *Chan et al. 2019 [27]* | Methamphetamine | 34 | * | Core review | |
| | *Kishi et al. 2013 [32]* | Cocaine, amphetamine | 14 | 741 | Supplementary review | |

*(Continued)*

**Table 2.** (Continued)

| Intervention | Review | Substance(s) | # RCTs | # Participants | Critical Assessment | Evidence Statement |
|---|---|---|---|---|---|---|
| **Anticonvulsants** | *Minozzi et al. 2015[33]* | Cocaine | 20 | 2068 | Core review | Sufficient evidence from reviews to discount the effectiveness of anticonvulsants for stimulant use disorder |
| | *Chan et al. 2019 [25]* | Cocaine | 48 | * | Core review | |
| | *Alvarez et al. 2010[34]* | Cocaine | 15 | 1236 | Supplementary review | |
| | *Chan et al. 2019 [27]* | Methamphetamine | 34 | * | Core review | |
| ➤ Topiramate[1] | *Singh et al. 2015 [35]* | Cocaine | 5 | 519 | Supplementary review | |
| | *Chan et al. 2019 [25]* | Cocaine | 48 | * | Core review | |
| | *Chan et al. 2019 [27]* | Methamphetamine | 34 | * | Core review | |
| **Psychostimulants** | *Castells et al. 2016[36]* | Cocaine | 26 | 2366 | Core review | Insufficient evidence from reviews to either support or discount the effectiveness of psychostimulants for stimulant use disorder |
| | *Chan et al. 2019 [25]* | Cocaine | 48 | * | Core review | |
| | *Perez-Mana et al. 2013[49]* | Amphetamine | 11 | 791 | Core review | |
| | *Chan et al. 2019 [27]* | Amphetamine | 34 | * | Core review | |
| | *Bhatt et al. 2016 [37]* | Amphetamine | 17 | 1387 | Supplementary review | |
| | *Perez-Mana et al. 2011[38]* | Cocaine, methamphetamine | 29 | 2357 | Supplementary review | |
| ➤ Modafinil[1] | *Sangroula et al. 2017[39]* | Cocaine | 11 | 896 | Supplementary review | |
| ➤ Methylphenidate[1] | *Dursteler et al. 2015[40]* | Cocaine | 5 | 363 | Supplementary review | |
| | *Chan et al. 2019 [27]* | Amphetamine | 34 | * | Core review | |
| **Opioid Agonists[2]** | *Castells et al. 2009[41]* | Cocaine | 37 | 3029 | Supplementary review | Insufficient evidence from reviews to either support or discount the effectiveness of OAT for cocaine use disorder |
| **NAC** | *Echevarria et al. 2017[42]* | Cocaine | 6 | 384 | Supplementary review | Insufficient evidence from reviews to either support or discount the effectiveness of NAC for cocaine use disorder |

[1] Indicates sub-type of intervention within the above class.

[2] Data reflects concurrent opioid-stimulant use only.

*Chan et al. total participants stratified by intervention not reported [25, 27]. CM, Contingency Management; CBT Cognitive Behavioural Therapy; NAC, n-acetylcysteine.

Psychosocial interventions were compared to standard care or to other psychosocial interventions. The study and population characteristics are summarized in Table 2, and a more detailed outline of study demographics and key results are summarized in S1 Table.

## Intervention and effects

**Study quality.** Of the 29 reviews, 10 were appraised as 'core' reviews and the remaining were considered 'supplementary'. Findings are summarized in Table 2, including systematic

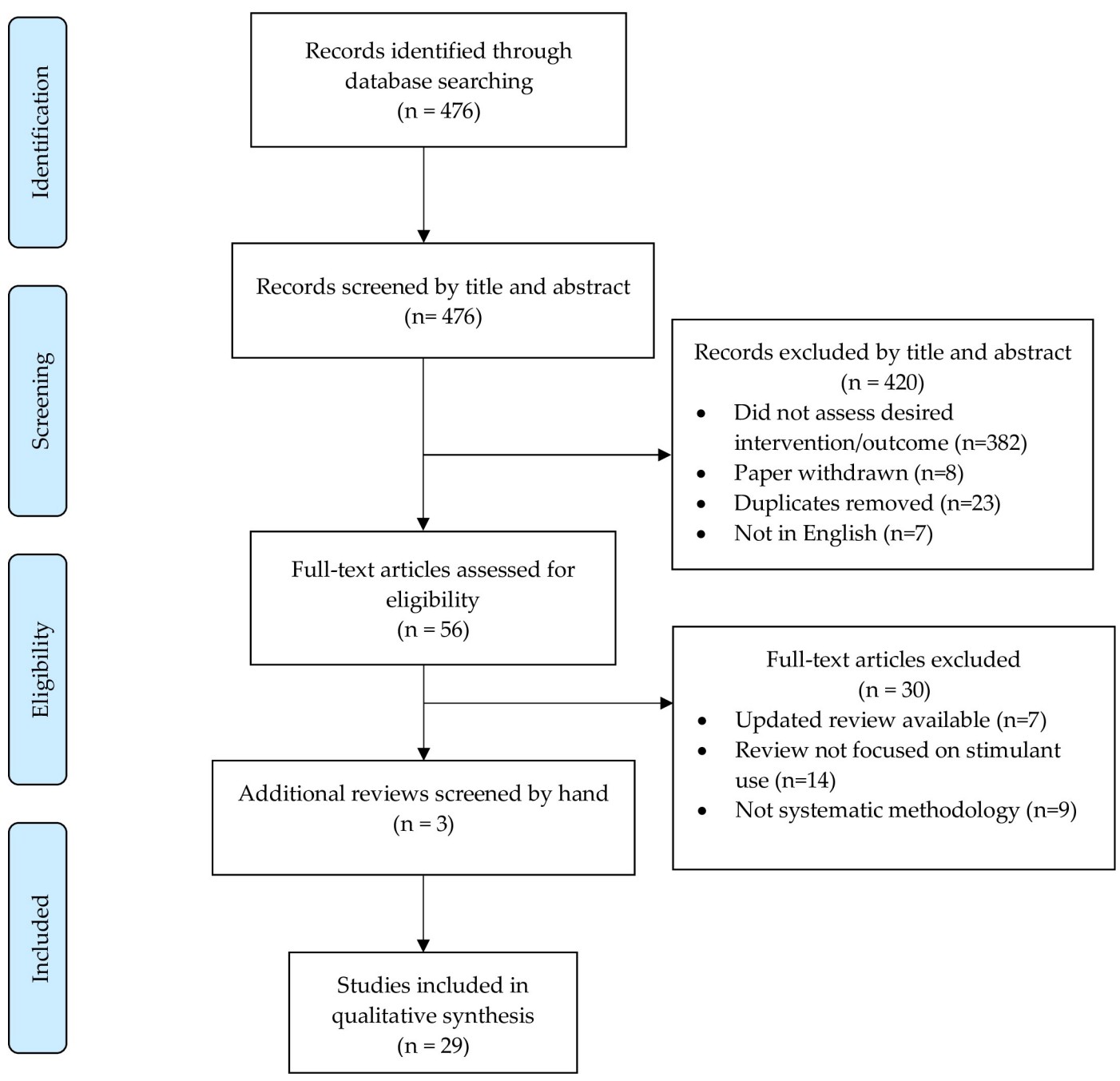

**Fig 1. Flow diagram of the selection and review process for review articles.**

reviews and meta-analyses assessed, number of randomized controlled trials in each, and our statement of evidence. A complete list of pharmacological interventions included in the reviews for this paper, stratified by drug type and number of trials, are listed in S3 Appendix. A complete summary of population characteristics and key findings of each review can be found in S1 Table.

## I. Behavioural interventions

**Contingency management.** Contingency Management (CM) involves participants receiving something of value such as a gift card, voucher or chance to win a prize as a reward for the achievement of a specific and measurable desired behaviour, most commonly a negative urine drug test for stimulants when implemented for the treatment of stimulant use disorder [20, 43]. Six trials have been included in this review assessing CM's efficacy in the treatment of stimulant use disorder and have found consistently positive results. De Crescenzo et al. was evaluated as core evidence, with the remaining five reviews evaluated as supplementary evidence. Pooled data demonstrates significant benefit across studies. CM has been evaluated both independently and in combination with psychotherapy and pharmacotherapy.

a. *Contingency management alone.* De Crescenzo et al. included 50 RCTs evaluating 12 psychosocial interventions for the treatment of stimulant use disorder, including 15 RCTs that compared CM alone to treatment as usual (TAU). A significant benefit was found for the outcomes of abstinence at 12 weeks (Odds Ratio [OR] 2.29, 95% Confidence Interval [CI] 1.62, 3.24), abstinence at the end of treatment (OR 2.22, 95% CI 1.59, 3.10), dropout at 12 weeks (OR 1.39, 95% CI 1.09, 1.78), and dropout at the end of treatment (OR 1.41, 95% CI 1.10, 1.82), but the effect was not sustained at longest follow up (OR 1.10, 95% CI 0.83, 1.46) [15].

Schumacher and colleagues assessed the efficacy of CM in comparison to day treatment programs in Alabama for the management of stimulant use disorder. Their meta-analysis of four successive RCTs demonstrated a significant benefit of contingency managed housing interventions, specifically abstinence-contingent housing and/or employment, among people who use crack cocaine and are homeless. In the primary outcome of abstinence at two months, the contingency management only group had a higher prevalence of cocaine negative urine drug test (prevalence: 0.71, standard error [SE]: 0.029) versus no treatment (prevalence: 0.46, SE: 0.036), or day treatment only (prevalence: 0.50, SE: 0.023). Of note, the addition of day treatment programs to CM (prevalence: 0.75, SE: 0.019) showed no added benefit compared to CM alone [16].

b. *Contingency management versus cognitive behavioural therapy.* When compared to Cognitive Behavioural Therapy (CBT), CM was found to have more immediate, but possibly shorter-term benefit [17, 18]. De Crescenzo et al. found CM alone as well as CM in combination with CBT to be superior to CBT alone in the outcome of abstinence at the end of treatment (OR 1.88, 95% CI 1.52, 2.85, and 2.08, 95% CI 1.28, 3.33, respectively). Of note CM compared to CBT at 12 weeks of treatment, and at longest follow up were not reported [15]. Farronato et al. found that CM alone resulted in reduced cocaine use during active treatment in all eight trials included in the study. Additive effects of CM and CBT were found in two of the five trials included [17]. CBT demonstrated less reliable benefit with no positive effect during active treatment, but showed delayed positive results in three out of five trials.

Similar results have been demonstrated for methamphetamine use disorder. Lee et al. included 12 RCTs of which four trials compared CM to CBT and three trials compared CM to usual care. Significant benefit was found for CM across studies for the outcomes of treatment retention, methamphetamine- or amphetamine- negative urine samples, and continuous periods of abstinence [18].

c. *Contingency management in combination with community reinforcement approach.* De Crescenzo et al. found the combination of CM with CRA to be the only intervention that

increased the rate of abstinence at the end of treatment (OR 2.84, 95% CI 1.24, 6.51), at 12 week follow up (OR 7.60, 95% CI 2.03, 28.37) and at longest follow up (OR 3.08, 95% CI 1.33, 7.17). They found this combination to be more effective than CBT (OR 2.44, 95% CI 1.02, 5.88), non-contingent rewards (OR 3.31, 95% CI 1.32, 8.28), and 12-step programme plus non-contingent rewards (OR 4.07, 95% CI 1.13, 14.69). CM with CRA also demonstrated significant benefit in the outcome of participant dropout when compared to TAU both at 12 weeks (OR 3.92, P < 0.001) and at the end of treatment (OR 3.63, P < 0.001) [15]. Schierenberg and colleagues evaluated CM in conjunction with standard treatment options for cocaine dependence, including CBT, pharmacotherapy and community reinforcement programs (CRA) [19]. The review included 16 studies and found that the addition of CM to CRA resulted in significant increase in cocaine abstinence and treatment retention. Roozen et al. included 11 RCTs and compared CM to CRA. The review found the addition of abstinence-contingent incentives (CM) to CRA to be superior to CRA alone in the treatment of cocaine dependence [20].

Of note, CRA was never explicitly evaluated in these reviews as it was only included as a comparison group. As a result, we are unable to assess its effectiveness alone based on current evidence.

*d. Contingency management in combination with pharmacotherapy*. Schierenberg et al. examined the effect of combining CM with pharmacotherapy for treatment of cocaine use disorder. CM had an additive effect when combined with pharmacological interventions, with a number needed to treat (NNT) between three and eight [19]. In a systematic review and meta-analysis by Castells et al. (2009) focusing on comorbid opioid and cocaine use disorder, the addition of contingency management for cocaine use to opioid agonist therapy was found to improve sustained cocaine abstinence [RR 3.11, 95% CI 1.80, 5.35] [44]

In summary, CM has been shown to have a significant benefit in treatment of both cocaine and stimulant use, and there may be an additive effect of CM when combined with both pharmacotherapy and other non-pharmacological interventions.

**Cognitive behavioural therapy.** CBT involves psychotherapy aimed at modifying a patient's thoughts and behaviours in order to reduce stimulant use. Its effectiveness in treating other substance use disorders has been well documented, however evidence is limited for its efficacy in treatment of stimulant use disorder [21]. The review by De Crescenzo et al. included seven RCTs comparing CBT to TAU. A significant benefit was found in the outcomes of dropout at 12 weeks (OR 1.42, 95% CI 1.05, 1.93), and dropout at the end of treatment (OR 1.47, 95% CI 1.08, 2.00). However, no significant benefit was identified for abstinence at 12 weeks, at the end of treatment, or at longest follow up [15].

A systematic review by Harada et al. evaluating the effectiveness of CBT for amphetamine-type stimulant use disorder included two RCTs and was unable to determine the effectiveness of CBT due to poor quality data [21]. There are limited available systematic reviews and meta-analyses focusing on CBT for the treatment of stimulant use disorder.

**Acupuncture.** Acupuncture has been explored for its use in a wide range of substance use disorders [45]. Mills et al. examined its use in treating cocaine dependence in a systematic review of nine RCTs including 1747 participants. Biochemical confirmation of abstinence was analyzed from seven studies and no significant benefit was found for acupuncture compared to controls. (OR: 0.76, 95% Confidence Interval [CI]: 0.45, 1.27). However results were weakened by large loss to follow up [22].

Gates and colleagues conducted a systematic review and meta-analyses of seven studies comparing auricular acupuncture to sham acupuncture or no acupuncture among 1,433

participants who used cocaine or crack cocaine. No difference was found between sham acu-puncture and auricular acupuncture with respect to biochemically validated cocaine use in either short-term (n = 3, Relative Risk [RR]: 1.01, 95% CI: 0.94, 1.08) or long-term (n = 1 RR: 0.98, 95% CI: 0.89, 1.09) follow up. Furthermore, no difference in self-reported cocaine use was found between acupuncture and no acupuncture groups in short term (n = 2, RR: 1.09, 95% CI: 0.71, 1.69) or long term (n = 2, RR: 1.04, 95% CI: 0.76, 1.43) assessment [23]. Results of the meta-analysis are limited by poor methodological quality of primary studies and small sample sizes.

Overall, results from several studies have shown no benefit in the use of acupuncture for treatment of stimulant use disorder.

## II. Pharmaceutical interventions

**Antidepressants.** Cocaine use increases availability of dopamine, serotonin and nor-adrenaline acutely, however results in down-regulation of these monoamine systems with chronic use. As a result, extensive investigation has been done into the use of antidepressants for treatment of stimulant use disorder [24]. A Cochrane review by Pani and colleagues included 37 randomized controlled trials covering a range of antidepressants, most commonly desipramine, fluoxetine and bupropion, and did not demonstrate benefit for cocaine depen-dence. Although there were positive results reported for mood-related outcomes consistent with the primary effects of antidepressants, these did not appear to affect outcomes directly related to cocaine abuse and dependence, including dropout (RR: 1.03, 95% CI: 0.93, 1.14) and abstinence from cocaine use (RR: 1.22, 95% CI: 0.99, 1.51 [24].

The recent systematic review and meta-analysis of pharmacotherapy for cocaine use disor-der by Chan et al. found no effect of antidepressants on rates of abstinence in a meta-analysis of ten RCTs [n = 1226, RR 1.27 CI 0.99, 1.63], or retention in treatment (33 RCTs, n = 2918, RR 0.95 CI 0.87, 1.03). Low quality evidence did support antidepressants in the outcomes of lapse (2 RCTs, n = 116, RR 0.79, CI 0.62, 1.00) and relapse (RR 0.74, 95%CI 0.57, 0.96) [25].

Torrens et al. stratified results by the presence of comorbid depression as well as by selective serotonin reuptake inhibitor (SSRI) versus non-SSRI medication among participants with cocaine dependence. Findings did not support the use of antidepressants in either group. In those without comorbid depression (14 RCTs), a statistically significant difference was noted with respect to reduction in cocaine use between SSRI (OR: 0.5, 95% CI: 0.22, 1.13) and non-SSRI (OR: 1.85, 95% CI: 1.06, 3.22) antidepressant medications, with results favouring the use of non-SSRI medications. In those with comorbid depression (five RCTs), meta-analysis showed no significant reduction in cocaine use for SSRI (OR: 0.54, 95% CI: 0.74, 7.30) or non-SSRI (OR: 2.32, 95% CI: 0.74, 7.30) antidepressants [26]. However, this study was categorized as a supplementary review due to limited sample size and low quality of individual RCTs.

A systematic review and meta-analysis by Chan et al. focusing on pharmacotherapy for methamphetamine or amphetamine use disorder found predominantly negative results for the use of antidepressants. No difference was identified for the outcome of abstinence, (4 RCTs, n = 590) or retention in treatment (6 RCTs, n = 831). For the outcome of reduction in use, authors reported mixed results; only one RCT (n = 60) demonstrated reduction in use when treated with mirtazapine, as measured by urinalysis [27].

In summary, review of antidepressant medication for the treatment of stimulant use disor-der has shown no significant benefit, although superiority of non-SSRI over SSRI antidepres-sant medications was noted in supplementary evidence.

**Disulfiram.** Disulfiram is FDA-approved for treatment of alcohol use disorder, but has also been explored for the treatment of other substance use disorders. This is because

disulfiram has been posited to act not only to inhibit aldehyde dehydrogenase, but as a broader enzyme inhibitor, including dopamine-beta-hydroxylase. Increased dopamine levels in meso-limbic circuits with the use of disulfiram may counter the depletion that is caused by chronic cocaine use [28]. A Cochrane review by Pani and colleagues examined seven studies with 492 participants for the use of disulfiram in the treatment of cocaine dependence and found mixed results. When compared to placebo, one out of four studies reported positive results for cocaine use, as measured by number of weeks of abstinence (weighted mean difference [WMD]: 4.50, 95% CI: 2.93, 6.07). No significant results were found for dropout (n = 2, RR: 0.82, 95% CI: 0.66, 1.03). When compared to naltrexone, no significant difference was noted in dropout rate (n = 3, RR: 0.67, 95% CI: 0.45, 1.01), although one study reported statistically sig-nificant reduction in cocaine use as measured by urine sample (WMD: -23.50, 95% CI: -26.58, -20.42). Finally, when comparing disulfiram versus no pharmacological treatment, one study found a statistically significant increase in weeks of consecutive abstinence (WMD: 2.10, 95% CI: 0.69, 3.51), and one study found a statistically significant increase in number of subjects to achieve three consecutive weeks of abstinence (RR: 1.88, 95% CI: 1.09, 3.23) [28].

In summary, these studies found low quality evidence to support the use of disulfiram for treating cocaine dependence though conclusions are limited by the small sample size of ran-domized controlled trials to date.

**Dopamine agonists.**   It has been hypothesized that dopamine agonists such as levodopa, cabergoline and pramipexole may reduce cravings, risk for relapse and withdrawal symptoms by increasing dopaminergic transmission in the mesolimbic pathway [29]. However, we found that the literature does not support the use of dopamine agonists for the treatment of stimulant use disorder. A Cochrane review by Minozzi et al. included 24 studies and 2147 participants, comparing dopamine agonists versus placebo in treatment of cocaine use disorder. No improvement was found in any of the outcomes studied, including dropout (RR: 1.04, 95% CI: 0.94, 1.14), abstinence (RR: 1.12, 95% CI: 0.85, 1.47), severity of dependence, and adverse events (RR: 1.27, 95% CI: 0.66, 2.44) [29].

**Antipsychotics.**   In contrast to dopamine agonists, antipsychotic medications block dopa-mine receptors and have been hypothesized to be effective by counterbalancing the increased dopamine neurotransmitter release from stimulant use [30]. The effects of atypical antipsy-chotics on the serotonergic system have also been proposed as an alternative mechanism of action for the treatment of stimulant use disorder [30]. A 2016 Cochrane review by Indave and colleagues included 14 studies and 719 participants with cocaine dependence. Antipsychotic medications were compared to placebo (11 studies) or to a different antipsychotic medication (three studies). The only primary outcome for which a significant difference was reported was reduction in study dropout compared to placebo in eight studies, with moderate quality of evi-dence (RR: 0.75, 95% CI: 0.57, 0.97). For all other primary outcomes including number of par-ticipants using cocaine during treatment, continuous abstinence, side effects, and cravings, no significant differences were reported, with low quality of evidence. Major biases reported were attrition bias in 40% of studies, and selection bias resulting from low quality reporting [30].

Chan et al. found predominantly negative results for the use of antipsychotics in the treat-ment of cocaine use disorder, with no difference reported in the outcome of abstinence (1 sys-tematic review of 3 RCTs, n = 139, RR 1.30 95%CI 0.73, 2.32), reduction in use, lapse, or relapse. The aforementioned systematic review by Indave et al. was also included in the review by Chan et al. and reported positive results for retention in treatment [25].

Alvarez and colleagues conducted a meta-analysis of 12 randomized, double-blind placebo controlled clinical trials with 681 patients focusing on the use antipsychotics for cocaine use disorder, which was included as supplementary evidence. Results included 48% loss to follow

up, no significant reduction in cocaine use (WMD: 0.01, 95% CI: -.12, 0.13), and no improvement in treatment retention (RR: 0.91, 95% CI: 0.82, 1.02) [31].

The use of antipsychotics for treatment of methamphetamine or amphetamine use disorder was evaluated in the systematic review and meta-analysis by Chan et al. No difference was reported for the use of aripiprazole for the outcomes of abstinence (1 RCT, n = 90), reduction in use (2 RCTs, n = 143), or retention in treatment (1 RCT, n = 53) [27].

Kishi et al. examined both cocaine and amphetamine use disorder, and found no difference between antipsychotic- and placebo- treated participants. The review was included as supplementary evidence. Outcomes reported included days of cocaine or amphetamine abstinence, addiction severity, craving, and mood related parameters [32].

In summary, antipsychotics had no superiority over placebo in treating stimulant use disorder but may result in greater study retention.

**Anticonvulsants.**   It is hypothesized that anticonvulsants may contribute to the treatment of stimulant use disorder by potentiating GABA inhibitory neurotransmission, thereby preventing the rise in dopamine that is caused by cocaine use. Conflicting evidence for the effectiveness of anticonvulsants for the treatment of stimulant use disorder has been reported in the literature [46–48]. Minozzi and colleagues conducted a Cochrane review of 20 studies including 2068 participants comparing anticonvulsants versus placebo for the treatment of cocaine dependence. There was no significant benefit demonstrated in any of the primary or secondary outcome measures, including dropout (17 studies, RR: 0.95, 95% CI: 0.86, 1.05), cocaine use (9 studies, RR: 0.92, 95% CI: 0.84, 1.02), and side effects as measured by the number of participants reporting at least one side effect (8 studies, RR: 1.39, 95% CI: 1.01, 1.9) [33].

Chan et al. found no effect of anticonvulsants in the treatment of cocaine use disorder. No difference was found for the outcomes of abstinence, reduction in use, retention in treatment, and withdrawal [25].

Alvarez and colleagues demonstrated insufficient evidence to support the use of anticonvulsant drugs in treatment of cocaine dependence. In this review of 15 RCTs including 1,236 participants, two outcome measures were assessed, with neither outcome demonstrating a significant improvement between the treatment and control groups: retention in treatment (15 studies, RR: 0.99, 95% CI: 0.90, 1.11), and cocaine use (13 studies, RR: 0.95, 95% CI: 0.85, 1.06) [34].

Chan et al. also assessed the use of anticonvulsants for the treatment of methamphetamine or amphetamine use disorder. No difference was reported for the outcomes of abstinence (1 RCT, n = 88) or retention in treatment (2 RCTs, n = 228) [27].

These studies are limited by a small number of trials for each anticonvulsant type and further clinical research in this area is warranted.

**Topiramate.**   Singh et al. included five peer-reviewed randomized controlled trials evaluating the use of topiramate in participants with cocaine dependence or cocaine use disorder in a systematic review and meta-analysis. No significant difference was reported for retention in treatment between topiramate-treated and control groups. Two out of five studies found improvement in continuous abstinence (RR: 2.43, 95% CI: 1.31, 4.53) and one study found a significant reduction in craving [35].

Chan et al. also reported positive results for topiramate in the outcome of cocaine abstinence. Results from two RCTs reported increased continuous abstinence for three or more weeks (RR 2.56, 95% CI 1.39, 4.73) [25].

Chan et al. also reported a reduction in use with topiramate for the treatment of amphetamine or methamphetamine use disorder (2 RCTs, n = 228) [27]. Although anticonvulsants for the treatment of stimulant use disorder have not shown it to be effective, data is limited on

individual medications in this drug class such as topiramate, and more study in this area may be of benefit.

**Psychostimulants.** It is hypothesized that psychostimulant treatment for stimulant use disorder may be effective by substituting a slow acting drug with a similar mechanism of action, thereby reducing withdrawal symptoms and cravings [36]. Eight reviews and meta-analyses assessing the use of psychostimulants for treatment of stimulant use disorder have been included in this review: four were appraised as core [25, 27, 36, 49], and four have been included as supplementary evidence[37–40]. Reviews focused on a single medication have been included as sub-categories.

The 2016 Cochrane review by Castells et al. included 26 studies with 2366 participants, examining nine drugs. Psychostimulants improved sustained cocaine abstinence (RR: 1.36, 95% CI: 1.05, 1.77) and NNT = 14, particularly for bupropion and dexamphetamine, but did not reduce use among participants actively using cocaine (standardized mean difference [SMD]: 0.16, 95% CI: -0.02, 0.33). Of note this review included all medications with some psychostimulant effect or those metabolised to a psychostimulant, including bupropion. There was moderate quality of evidence that psychostimulants did not improve treatment retention. There was no significant difference in adverse events between psychostimulants and placebo. In participants who used both opioids and stimulants and treated with methadone, sustained cocaine and heroin abstinence increased with psychostimulant treatment compared to placebo [36]. Attrition bias was a major limitation to these trials and further investigation into psychostimulant substitution treatment is encouraged.

The systematic review and meta-analysis by Chan et al. assessed the efficacy of psychostimulants for treatment of cocaine use disorder. Positive results for the outcome of abstinence were reported from the systematic review by Castells et al. No difference was reported for the outcomes of any cocaine use or retention in treatment [25].

A Cochrane review of 11 studies including 791 participants was conducted by Perez-Mana et al. examining the use of psychostimulants in the treatment of amphetamine use disorder. There was no significant improvement in sustained abstinence as measured by negative urinalysis over three consecutive weeks (RR: 1.12, 95% CI: 0.84, 1.49), or retention in treatment (RR: 1.01, 95% CI: 0.9, 1.14). The medication was well tolerated and adverse events were rare [49]. Data focusing specifically on amphetamine dependence is limited, and further investigation including larger and longer trials are needed to establish the utility of psychostimulants in this area.

Chan et al. found limited evidence for the use of psychostimulants for methamphetamine or amphetamine use disorder. No difference was reported for the outcomes of abstinence (1 SR, 2 RCTs, OR 0.86, 95%CI 0.46, 1.61) or retention in treatment (1 SR, 11 RCTs, OR 1.11, 95% CI 0.86, 1.44). Mixed results were reported for the outcome of reduction in use, with 8 RCTs reporting no difference, two RCTs with a positive effect of methylphenidate, and two RCTs with no effect [27].

A recent systematic review and meta-analyses by Bhatt et al. included 17 studies with 1387 participants, assessing psychostimulants for treatment of methamphetamine dependence. It was included in this review as supplementary evidence due to small sample size. The main outcome measure was sustained abstinence, for which no effect was reported (OR: 0.97, 95% CI: 0.65, 1.45). However only five studies were included in this subgroup analysis. Authors found no effect on treatment retention, but a small subgroup analysis suggested improvement with longer duration of psychostimulant treatment [37].

Perez-mana and colleagues conducted a systematic review in 2011 assessing indirect dopamine agonists (IDA) in treating psychostimulant dependence, including both cocaine and methamphetamine. Although this review included interventions with a broad range of

mechanisms of action, positive results were found primarily in those studies assessing psychostimulant medication for cocaine dependence. IDAs considered to have psychostimulant effect included bupropion, dexamphetamine, mazindol, methamphetamine, methylphenidate, and modafinil. The analysis included 29 studies with 2,467 participants. Primary outcome measures assessed were stimulant abstinence and retention in treatment. Subgroup analysis of IDAs with psychostimulant effect showed minor but statistically significant positive results in the outcome of abstinence (14 studies, SMD: 0.22, 95% CI: 0.05–0.40), but no improvement in treatment retention (20 studies, RR: 1.00, 95% CI: 0.92, 1.09) [38]. Similar results were reported by Castells et al. for bupropion, dexamphetamine, disulfiram and mazindol as adjunct therapy to opioid agonist therapy in dual opioid-cocaine use disorder, with an increase in sustained cocaine abstinence (RR 1.44, 95% CI 1.05, 1.98) [36].

**Modafinil.** A 2017 systematic review and meta-analysis of 11 RCTs by Sangroula et al. assessed the efficacy of modafinil for cocaine dependence. The primary outcome of treatment retention was not significantly different between the treatment and placebo groups, but significant positive results were found for the secondary outcomes of number of cocaine non-use days (SMD: -1.294, 95% CI: –2.572, 0.017) and proportion of negative urine samples (SMD: -0.633, 95% CI: –1.248, 0.018). No significant differences were found with respect to safety and adverse events [39].

**Methylphenidate.** A review of the literature published by Dursteler et al. in 2015 concluded that methylphenidate treatment of up to 90 mg per day for cocaine-dependent participants is safe, however did not significantly reduce cocaine use. These results were generalizable to adults with or without ADHD and with or without concurrent opioid agonist treatment [40]. However, in the Chan et al. review of treatment of amphetamine and methamphetamine use disorder, two RCTs found a positive effect of methylphenidate on reduction in use, one trial reporting 6.5% versus 2.8% mean proportion amphetamine-negative urine samples in an intent-to-treat analysis (adjusted odds of positive urine sample 0.46, 95% CI 0.26, 0.81, p = 0.008), and another reporting 23% versus 16% amphetamine-negative urines (n = 54, p 0.047) [27].

In summary, review of the evidence of psychostimulant treatment of cocaine as well as amphetamine and methamphetamine dependence has shown some promising results, although further investigation is required for more definitive clinical evidence.

**Opioid agonist.** Opioid agonist therapy (OAT), including buprenorphine-naloxone and methadone, are first and second line treatments for opioid use disorder [50]. A systematic review and meta-analysis by Castells and colleagues published in 2009 assessed the effectiveness of opioid agonists in the treatment of concurrent opioid and stimulant use disorder. Results included 37 studies, with 3,029 participants who used both cocaine and heroin. Higher doses of OAT had no effect on cocaine abstinence compared to lower doses. When compared to buprenorphine, methadone demonstrated greater improvement in cocaine abstinence (RR: 1.63, 95% CI: 1.20, 2.22). Adjunctive treatment options including indirect dopamine agonist therapy and CM were found to significantly improve cocaine abstinence [41]. Of note, no studies compared OAT to placebo, and it is difficult to decipher the direct effect of OAT on cocaine use from the benefit of parallel reduction in heroin use, in the context of dual heroin-cocaine dependence.

**N-acetylcysteine.** N-acetylcysteine (NAC) is used primarily to treat acetaminophen overdose, and as a mucolytic [42]. It has also been hypothesized to be of use in treating cocaine use disorder by restoring glutamate transporter-1 (GLT-1), by clearing excess glutamate from the extrasynaptic space, thus restoring glutamatergic homeostasis that is impaired with chronic stimulant use [42, 51]. A recent systematic review by Nocito Echevarria et al. summarized the current data from six human trials and 16 animal studies on the use of NAC for cocaine use

disorder. Preliminary results of this review reported significant reduction in craving, "cocaine-cue viewing-time" and "cocaine-related spending" in four clinical trials. Side effects were reported as mild and did not vary significantly between doses [42]. Results were included in this review as supplementary evidence due to limited human data and qualitative methodology.

## Discussion

This review synthesized the findings from 26 systematic reviews related to the psychosocial and pharmacological interventions for SUD. Although stimulant use remains a prominent issue worldwide, our review suggests that evidence-based treatment options are limited. We observed the strongest body of evidence exists for contingency management. The pharmacological intervention that shows the most promise is psychostimulant agonist therapy. Some positive results have also been reported for OAT, NAC, Disulfiram, and antidepressants for methamphetamine use. All other interventions reviewed here, including dopamine agonists, antipsychotics, anticonvulsants, and acupuncture, have found predominantly negative results.

For contingency management (CM) programs, we found consistently positive results across five systematic reviews demonstrating their effectiveness compared to treatment as usual, as well as other interventions, including community reinforcement, pharmacotherapy, and CBT. Furthermore, CM may be supplemented with CBT or CRA to ensure both short and long term success, given the demonstrated delayed benefit of CBT, and additive effects of CM with CRA [15, 17, 18, 52]. Despite these positive results, CM programs are rarely implemented, and questions remain about the long-term benefits associated with CM interventions. Barriers that have been identified in the literature include treatment providers viewing programs as too costly, difficult to implement, or not aligning with political or philosophical values [53]. A qualitative assessment of treatment providers' opinion on barriers to incentive based programs found many view the intervention positively, but that cost and training of providers was a significant barrier [53]. Randomized controlled trials, however, have demonstrated the cost effectiveness of contingency management programs [54, 55]. In a trial by Peirce et al., the intervention was successful at just $1.46 per participant per day, rather than an average cost of $120 per day that has been previously reported [54]. Given strong evidence for its effectiveness, and in the absence of other similarly efficacious interventions, efforts to expand access to contingency management programs for stimulant use are warranted.

Psychostimulant treatment has a similar pharmacological rationale as other evidence-based substance treatments like nicotine replacement and opioid agonist therapies [56, 57]. The core review by Castells and colleagues found significant positive results for sustained cocaine abstinence and found positive, but insignificant results for reduction in use [36]. These results were reiterated in the core review by Chan et al [25]. Therefore, though we assessed the available evidence to date as insufficient to support or discount the use of psychostimulants for treatment of cocaine use disorder, this class of medication warrants further investigation. Chan et al. also reported positive results in the outcome of reduction in use for the use of methylphenidate for treatment of methamphetamine use disorder, and this area warrants further investigation [27]. Further evaluation of the outcomes of reduction in use as well as sustained abstinence would allow for better pooling of results and increase the quality of evidence available to recommend clinical practice. Evaluation of the effectiveness of psychostimulants should also include trials of longer duration, as the average length of trial in the review by Castells et al. (2016) was 12.6 weeks (range 6–24) which may be insufficient to achieve abstinence. Furthermore, subgroup analyses addressing optimal dosing, as well as trials that evaluate combination pharmacotherapy may provide further insight into the effectiveness of the intervention.

Positive results have also been noted for CBT, OAT, NAC, disulfiram, and Antidepressants, though data is not sufficient to recommend their use due to limitations in data quality and sample size. The available data for OAT focuses on dual opioid-cocaine use disorders, which may be an important area of future study given the concurrent rise in opioid and cocaine use in recent years [6]. Castells et al. found a significant superiority of methadone when compared to buprenorphine in reduction of cocaine use, however it is important to note that OAT was not compared to a control group, and no dose response was observed for the methadone group [41]. Further characterization of this finding as well as evaluation of other OAT medications may help guide clinical practice in this area.

CBT has been studied both alone and in combination with other psychotherapy interventions for treatment of stimulant use disorder, with some positive results. De Crescenzo et al. found a reduction in participant dropout with CBT alone, however when combined with CM, CBT was found to have a more pronounced effect, including in the outcome of participant abstinence [15]. The available data for CBT in the treatment of stimulant disorder is limited, and more research is warranted to determine its clinical utility, focusing on a potentially more sustained effect when employed complementarily to contingency management psychosocial interventions, or pharmacologic treatment options.

The available data for the possible benefits of NAC is quite limited, with only six human trials included in the review by Nocito Echevarria et al. and further investigation is necessary to evaluate its clinical utility.

Similarly, for disulfiram, data supporting its clinical utility in treatment of stimulant use disorder is limited. The review by Pani et al. found positive results in reduction in use for one of four included RCTs [28]. However, concerns regarding the safety of Disulfiram, particularly in concurrent alcohol users may limit its potential for future research[58].

Current data does not support the use of antidepressants for treatment of cocaine use disorder, although with regards to amphetamine use there is insufficient evidence to discount its use. Due to predominantly negative results for the use of antidepressants for treatment of cocaine use disorder, it may be possible to extrapolate from this data for the treatment of amphetamine use disorder [24, 25, 27]. However, due to the unique mechanisms of action on presynaptic monoamine reuptake transporters, further research focused on the utility of antidepressants for amphetamine-type stimulants may be warranted.

It is important to note that the available literature in treatment for stimulant use disorder is primarily focused on cocaine use disorder, rather than amphetamine or methamphetamine, for which data is extremely limited. Although the mechanism of action of these substances are similar, there are important distinctions that should be addressed moving forward to expand research in this area, and when applying the evidence to clinical practice. Both cocaine and methamphetamine act to increase the availability of monoamines in the synapse, however cocaine acts as a reuptake inhibitor, whereas methamphetamine binds transporters at the presynaptic membrane and is exchanged to release more monoamine neurotransmitter into the synapse [59]. Methamphetamine use has been increasing in the United States since 2011, and is the most commonly identified substance associated with violent crime [6, 60]. Given the rising rates of methamphetamine use, its associated harms, and the differences in mechanism of action between stimulants, it is critical that future studies evaluate outcomes for both substances.

Another limitation of meta-analyses to date is the lack of standardized outcomes, making pooling of data difficult. Without evidence-based, standardized clinical trial outcomes for the treatment of stimulant use disorder, it will remain difficult to pool data and provide strong clinical recommendations. Long-term cessation of use has traditionally been the primary goal of substance use treatment and abstinence measures have been the most commonly

implemented standard outcomes in randomized controlled trials [61]. However, the definition of recovery may vary based on individual patient goals, by feasibility within the study time period, and may not always include abstinence [62]. In 2015, the Analgesic, Anesthetic, and Addiction Clinical Trial Translations, Innovations, Opportunities, and Networks (ACTTION) group made recommendations for meaningful indicators of treatment success in future clinical trials on stimulant use, including a focus on the validation of patient reported outcome measures of functioning and the exploration of patterns of stimulant abstinence that may be associated with physical and/or psychosocial benefits [61]. Future study may benefit from patient-oriented outcome measures, including psychosocial parameters such as improved social functioning, employment and acquisition of adequate housing.

It is important to note that the variability in treatment response among population subgroups is poorly addressed in the literature. Given the extensive variability in individual response to treatment, it is possible that an intervention appraised as 'sufficient evidence to discount the effectiveness' or 'insufficient evidence to support or discount its effectiveness' may have significant benefits for some patients or patient subgroups. In studies that performed subgroup analyses, including but not limited to: age, sex, severity of substance use disorder and comorbid substance use disorders, meaningful results were limited. Future research would benefit from identifying those subgroups that may be more likely to benefit from certain interventions.

Subgroup analyses by Bhatt et al. demonstrated increased retention in treatments of longer duration ($\geq$ 12 weeks)[37]. Stratification by treatment duration moving forward may aid in identifying optimal and minimum effective treatment durations. Castells et al. (2016) found that psychostimulants increased abstinence and reduced cocaine use in those studies in which Attention Deficit Hyperactivity Disorder (ADHD) was not an inclusion criteria, which may be an important replicate moving forward[36].

Finally, polysubstance use is common [63, 64]. Recent data demonstrates a rise in concurrent amphetamine and opioid use, and the role of stimulant use in the overdose epidemic remains poorly defined [65]. Several of the included reviews identified that many participants suffered from polysubstance use disorder, however the effect of polysubstance use on treatment efficacy was rarely addressed. Future research would benefit from identifying the efficacy of interventions for stimulant use disorder specifically in the context of polysubstance use.

Alternate pharmacotherapy for stimulant use disorder has been proposed in the literature, with limited RCTs available and no systematic review to date. These include naltrexone, for which several RCTs have been conducted with some positive results, [66–71] and buspirone [72]. Furthermore, novel therapies have been evaluated in pre-clinical studies including ibogaine, lobeline, TV-1380, and vaccines to combat substance use disorder. These alternative therapies are beyond the scope of this review, although may warrant further investigation [73–76].

This review has several limitations. Notably, attrition bias was common across several studies, reducing the power of results in intention to treat analysis. Several factors may contribute to participant drop-out including heavy substance use, financial and transportation barriers, and ambivalence toward abstinence [77]. Furthermore, due to the systematic review of reviews methodology, there is primary data published after the included studies that will not contribute to our findings. Our search was limited to English language literature and as a result we may not have included some important data. Finally, our assessment of quality of evidence was based on the methods of each review, which may not be adequate. There, we did not critically appraise the primary literature itself, rather used the assessment done by the authors of those reviews included in this paper. This is to our knowledge however, the first 'review of reviews' to synthesize the available literature for treatment of stimulant use disorder.

## Conclusions

This review synthesized the evidence to date for treatment of stimulant use disorder, including both pharmacological and psychosocial interventions. Despite the extensive amount of research in this area, little clinical application has resulted thus far. The strongest evidence-based approach for the treatment of stimulant use disorder at this time remains contingency management interventions. While treating stimulant use disorder with psychostimulants has shown some favourable results, high quality clinical trials and meta-analyses are needed to determine the clinical utility of psychostimulants and other pharmacotherapies to address the growing need for stimulant treatments.

## Supporting information

**S1 Appendix. PRISMA checklist.**
(PDF)

**S2 Appendix. Search strategy.**
(DOCX)

**S3 Appendix. Complete list of pharmaceutical interventions assessed in this review.**
(DOCX)

**S1 Table. Summary of population characteristics and key findings.**
(DOCX)

## Author Contributions

**Conceptualization:** Claire Ronsley, Rod Knight, Alex Walley, Nadia Fairbairn.

**Data curation:** Claire Ronsley, Nadia Fairbairn.

**Formal analysis:** Claire Ronsley, Seonaid Nolan, Nadia Fairbairn.

**Methodology:** Claire Ronsley, Jano Klimas, Nadia Fairbairn.

**Resources:** Nadia Fairbairn.

**Supervision:** Nadia Fairbairn.

**Writing – original draft:** Claire Ronsley.

**Writing – review & editing:** Seonaid Nolan, Rod Knight, Kanna Hayashi, Jano Klimas, Alex Walley, Evan Wood, Nadia Fairbairn.

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
