## [Decision Letter · Decision Letter 0]

1 Apr 2020

PONE-D-20-02936

Treatment of stimulant use disorder: a systematic review of reviews

PLOS ONE

Dear Dr. Fairbairn,

Thank you for submitting your manuscript to PLOS ONE. After careful consideration, we feel that it has merit but does not fully meet PLOS ONE’s publication criteria as it currently stands. Therefore, we invite you to submit a revised version of the manuscript that addresses the points raised during the review process.

The two reviewers addressed several major and minor concerns about your manuscript. Please revise your manuscript carefully.

We would appreciate receiving your revised manuscript by May 16 2020 11:59PM. To enhance the reproducibility of your results, we recommend that if applicable you deposit your laboratory protocols in protocols.io, where a protocol can be assigned its own identifier (DOI) such that it can be cited independently in the future. For instructions see: http://journals.plos.org/plosone/s/submission-guidelines#loc-laboratory-protocols

We look forward to receiving your revised manuscript.

Kind regards,

Kenji Hashimoto, PhD

Academic Editor

PLOS ONE

"This research was undertaken, in part, thanks to funding from a MSFHR/St. Paul’s Foundation Scholar Award which supports Dr. Nadia Fairbairn. Evan Wood is supported by a Tier 1 Canada Research Chair in Addiction Medicine. Kanna Hayashi is supported by a CIHR New Investigator Award (MSH-141971), a Michael Smith Foundation for Health Research (MSFHR) Scholar Award, and the St. Paul’s Hospital Foundation. Dr. Seonaid Nolan is supported by the MSFHR and the University of British Columbia’s Steven Diamond Professorship in Addiction Care Innovation. Rod Knight is supported by a Scholar Award from MSFHR. A European Commission grant (701698) and Canadian Institutes of Health Research (671 397968, 422332) grants support Dr. Klimas."

"The authors received no specific funding for this work."

5. Please upload a copy of Supporting Information Table S1-S2 which you refer to in your text on page 4 and 5.

Reviewers' comments:

Reviewer's Responses to Questions

**Comments to the Author**

1. Is the manuscript technically sound, and do the data support the conclusions?

Reviewer #1: Yes

Reviewer #2: Yes

2. Has the statistical analysis been performed appropriately and rigorously? 

Reviewer #1: N/A

Reviewer #2: N/A

3. Have the authors made all data underlying the findings in their manuscript fully available?

Reviewer #1: Yes

Reviewer #2: Yes

4. Is the manuscript presented in an intelligible fashion and written in standard English?

Reviewer #1: Yes

Reviewer #2: Yes

5. Review Comments to the Author

Reviewer #1: This is a review of the manuscript titled “Treatment of stimulant use disorder: A systematic review of reviews.” The manuscript is well written and clear. The majority of my comments are aimed at improving the manuscript as I did not identify major insurmountable flaws.

Major Comments

1. Abstract – The Abstract contradicts the content of the manuscript. Table 2 identified CM (sufficient evidence) and OAT (tentative evidence) as the only two categories with support for efficacy. There were several categories of potential pharmacotherapies in which the evidence statement by the authors indicated there is insufficient evidence to support or discount a given class of medications (i.e., CBT, antidepressants for methamphetamine, disulfiram, psychostimulants (for methamphetamine or cocaine), and NAC). The remaining categories’ evidence statements tentatively or sufficiently discounted the given categories as potential treatments. The abstract correctly and explicitly describes CM as having sufficient supporting evidence. However, the abstract also states psychostimulants, NAC, and OAT had positive results and that all other treatment options are not supported; these statements (last 2 sentences in Results paragraph) directly contradicts evidence statements in Table 2. A similar contradiction is found in the Conclusions paragraph where the psychostimulant class is summarized as “demonstrating the most promise.”

2. Table 2 – Great table, yet I have some recommendations that would improve it further, enabling readers to make better use of your table and match it with the text.

a. I am aware of a few reviews that fell within the authors’ search dates and apparent search criteria, but was surprised to not find them listed, i.e., de Crescenzo et al 2018; Tardelli et al 2018; and Ballester et al 2017. Presumably they were excluded from this review of reviews, but I share these specific citations out of concern for their lack of inclusion.

b. Schumacher et al. 2007 is not contained in the References list.

c. Adding the Reference # for each of the citations in the table will greatly improve the usefulness of the table.

d. Highly recommend moving the Substance column to the second column position and the Review column to the third position. Then, ensure all of the cocaine rows are together and the methamphetamine rows are together within one Intervention row. E.g., rearrange the 5 rows in CM so cocaine is all together.

e. The order of the rows (at the level of the citations) does not match the order the manuscript describes them, making reading the manuscript with the table side-by-side a bit cumbersome. Recommend reordering the rows to match the text, as the text does flow well.

f. Given the third aim of the paper is to guide future research, I was surprised to find the Discussion addressed some, but not all, of the categories with evidence statements of insufficient evidence to support or discount. i.e., Psychostimulants and NAC have this evidence statement and are discussed in Discussion, but antidepressants for methamphetamine and disulfiram have the same evidence statement but are ignored in the Discussion.

3. Similar to comment 2.e., the order within the Psychostimulants (Line 451) section is particularly jumpy…jumps back and forth between amphetamine and cocaine within the text itself. I believe this review is well-positioned by continually discussing results by cocaine and by amphetamines/methamphetamine separately. The two use disorders just need to be consistently grouped together to help the reader assimilate the information.

4. Appendix 1 – Most, but not all, of the citations in the rows of Table 2 are found in this appendix. Why are not all represented? This seems problematic. Secondly, recommend Appendix 1 be reordered to match Table 2.

Minor Comments

5. Line 94 – 96: This sentence does not fit well where it is currently placed.

6. Line 133: “DSM-V” should be abbreviated “DSM-5”.

7. Line 161 – 163: Sentence on these lines interrupt the flow of the content above and below it. Recommend moving it to the end of the next paragraph.

8. Line 174 – 175: Reports discrepancies were resolved by third independent reviewer. Can the authors add information/data about the extent of discrepancies.

9. Line 221 – Schumacher not in References. Appears to be incorrectly cited as #15 on Line 229. Also, why is this (potentially) primary study (Schumacher) discussed in detail in this paragraph? Is it meant to be an exemplar and if so, why this one?

10. Line 254 – citation #14 should be #15.

11. Line 271 – replace “were” with “was”.

12. Ensure abbreviations “WMD” and “SMD” are spelled out at first use.

13. Line 586 – citation #55, is this correct? Should it be #39?

14. Line 595 – which Castells et al. are the authors referring to here? There are 3 Castells et al.

15. Ensure References contain all required elements. I happened to notice citation #55 is missing the year and issue details.

Reviewer #2: This is a timely review of effectiveness of treatment for stimulant use disorder. This topic is of much interest and recent meetings have focused on what we know and what work needs to be done to improve treatment options and effectiveness. This article provides an organized evaluation of currently available information through published systematic reviews.

With that said, three very important topics of discussion are missing from this paper to make it most relevant to the current discussion and should be incorporated in a way that put the findings in context of the data and the authors interpretations and conclusions:

1) The definition of "recovery" may no longer include abstinence. While the authors point out the limitation of standard outcome measures, this needs to be specifically addressed with current views and how it impacts the interpretation of these findings.

2) The assessment of treatments assumes that one type of therapy is effective for all patients. This is a serious issue considering we know that not all patients respond in the same manner. More information is desired for the populations treated in these studies, perhaps adding some information to Table 2. The study population is included is some of the narratives but not as an overall important piece of the evaluation. Understanding patient characteristics that improve outcomes with certain therapies is critical to improving treatment. What is known about factors or drivers that impact successful therapy? And how does this impact your findings?

3) Polysubstance use is the norm and is absent from this discussion and how it effects the interpretation of these findings.

General comments:

1) The citations used in the introduction are outdated. Is there a reason why more recent data are not cited? This is important as substance abuse patterns have changes in the past few years.

2) Not all references are in the correct format. Please correct.

6. PLOS authors have the option to publish the peer review history of their article (what does this mean?). If published, this will include your full peer review and any attached files.

Reviewer #1: No

Reviewer #2: No

---

## [Author Response · Author response to Decision Letter 0]

16 May 2020

RESPONSE TO REVIEWER’S COMMENTS:

 *Any new text in the revised manuscript has been noted below in bold font.

Editor:

1. In accordance with PLOS ONE’s style requirements, we have updated the manuscript title and references, as well as the file names.

2. Description of funding information has been removed from the manuscript in accordance with PLOS requirements. We would like to update our Funding Statement as follows:

"This research was undertaken, in part, thanks to funding from a MSFHR/St. Paul’s Foundation Scholar Award which supports Dr. Nadia Fairbairn. Evan Wood is supported by a Tier 1 Canada Research Chair in Addiction Medicine. Kanna Hayashi is supported by a CIHR New Investigator Award (MSH-141971), a Michael Smith Foundation for Health Research (MSFHR) Scholar Award, and the St. Paul’s Hospital Foundation. Dr. Seonaid Nolan is supported by the MSFHR and the University of British Columbia’s Steven Diamond Professorship in Addiction Care Innovation. Rod Knight is supported by a Scholar Award from MSFHR. A European Commission grant (701698) and Canadian Institutes of Health Research (671 397968, 422332) grants support Dr. Klimas." 

3. As requested for PLOS submissions, we have updated ORCID ID information and validated it in Editorial Manager. 

4. Supporting Information files have been included at the end of the manuscript, and in-text citations have been updated in accordance with PLOS guidelines. 

5. Appendix S1 and S2 referenced on pages 4 and 5 of the manuscript have been updated and uploaded as Supporting Information. 

Reviewer #1: 

We thank Reviewer #1 for identifying discrepancies between the overview of included systematic reviews and statements of evidence presented in Table 2 and the Abstract and Conclusion sections of the manuscript. Discrepancies were correctly identified by the Reviewer with respect to psychostimulants, n-acetylcysteine (NAC) and opioid agonist therapy (OAT) interventions. We have made several changes to ensure the evidence statements in Table 2 align with the Abstract:

The summary of evidence for psychostimulants is also now aligned in the Abstract section with Table 2, which found insufficient evidence to support or discount its use. 

For NAC, we have removed the statement in the Abstract regarding positive results for NAC to align with the evidence statement in Table 2 which found insufficient evidence to support or discount its use. 

Regarding the evidence statement for OAT, the study reviewers re-appraised the included review by Castells et al. (2009) and have adjusted the evidence statement to be insufficient to support or discount its use (Table 2). The evidence statement for OAT was previously categorized as tentative evidence to support the effectiveness of the intervention. The study reviewers decided to adjust the evidence statement for OAT because the positive benefits for OAT in previous reviews are limited to a select subgroup of people with dual heroin-cocaine dependence, and only comparative evidence between OAT therapies were found, with no comparisons between OAT and placebo. We thank Reviewer #1 for bringing this discrepancy to our attention.

Accordingly, we have revised the results section of the Abstract to align with the evidence statements presented in Table 2, as follows:

 “Psychostimulant, n-acetylcysteine, opioid agonist therapy, disulfiram and antidepressant pharmacological interventions were found to have insufficient evidence to support or discount their use.” (Abstract, page 2)

We have also revised the conclusion statement regarding psychostimulants in the Abstract as follows:

“Although evidence to date is insufficient to support the clinical use of psychostimulants, our results demonstrate potential for future research in this area.” (Abstract, page 2)

We appreciate the suggestion from Reviewer #1 to align the presentation of evidence statements in Table 2 with the Results section of the manuscript to improve ease of interpretation of data for the reader. We have now arranged both the Results section and Table 2 by cocaine use first, followed by amphetamine/methamphetamine use. Additionally, core reviews are reported before supplementary reviews in the text. Table 2 now aligns with the order of presentation of interventions in the Results section of the manuscript and should be easier to follow for the reader.

Reviewer #1 also brought forward the names of several review papers on treatment of stimulant use disorder for further information regarding our decision not to include them in the current study.

Regarding the De Crescenzo et al. (2018) article, this review was a comparative efficacy and acceptability study of a range of psychosocial interventions for individuals with cocaine and amphetamine addiction. This study was initially identified using our search strategy and was excluded based on our criteria not to include reviews where the intervention was not clearly defined, due to the fact that this study primarily focused on comparative effectiveness of psychosocial treatments used in combination (Methods section, page 5). However, the reviewers scrutinized this study again and determined that it is possible to separate results by intervention in this study as per our inclusion criteria for the study, and have thus opted to include it in our review. We thank the Reviewer for their helpful feedback and feel the manuscript is strengthened by inclusion of this paper in our study. We have updated Table 2 accordingly, and have added additional text in the manuscript regarding findings from this review (Results, pages 15-17): 

“De Crescenzo et al. included 50 RCTs evaluating 12 psychosocial interventions for the treatment of stimulant use disorder, including 15 RCTs that compared CM alone to treatment as usual (TAU). A significant benefit was found for the outcomes of abstinence at 12 weeks (Odd’s Ratio [OR] 2.29, 95% Confidence Interval [CI] 1.62, 3.24), abstinence at the end of treatment (OR 2.22, 95% CI 1.59, 3.10), dropout at 12 weeks (OR 1.39, 95% CI 1.09, 1.78), and dropout at the end of treatment (OR 1.41, 95% CI 1.10, 1.82), but the effect was not sustained at longest follow up (OR 1.10, 95% CI 0.83, 1.46) (15).” (Results, page 15)

“De Crescenzo et al. found CM alone as well as CM in combination with CBT to be superior to CBT alone in the outcome of abstinence at the end of treatment (OR 1.88, 95% CI 1.52, 2.85, and 2.08, 95% CI 1.28, 3.33, respectively). Of note CM compared to CBT at 12 weeks of treatment, and at longest follow up were not reported (15).” (Results, page 15, 16)

“De Crescenzo et al. found the combination of CM with CRA to be the only intervention that increased the rate of abstinence at the end of treatment (OR 2.84, 95% CI 1.24, 6.51), at 12 week follow up (OR 7.60, 95% CI 2.03, 28.37) and at longest follow up (OR 3.08, 95% CI 1.33, 7.17). They found this combination to be more effective than CBT (OR 2.44, 95% CI 1.02, 5.88), non-contingent rewards (OR 3.31, 95% CI 1.32, 8.28), and 12-step programme plus non-contingent rewards (OR 4.07, 95% CI 1.13, 14.69). CM with CRA also demonstrated significant benefit in the outcome of participant dropout when compared to TAU both at 12 weeks (OR 3.92, P < 0.001) and at the end of treatment (OR 3.63, P < 0.001) (15).” (Results, page 16, 17)

“The review by De Crescenzo et al. included seven RCTs comparing CBT to TAU. A significant benefit was found in the outcomes of dropout at 12 weeks [OR 1.42, 95% CI 1.05, 1.93], and dropout at the end of treatment [OR 1.47, 95% CI 1.08, 2.00]. However, no significant benefit was identified for abstinence at 12 weeks, at the end of treatment, or at longest follow up (15).” (Results, page 18)

The Tardelli et al. (2018) paper provided an interesting review on contingency management (CM) combined with pharmacotherapy. We reviewed this paper again as it was identified as part of our search strategy and not included as it did not meet our inclusion criteria. It was a qualitative review rather than a systematic review and meta-analysis that did not meet our inclusion criteria due to its study design.

Similar to the Tardelli et al. (2018) paper, the Ballester et al. (2017) paper was reviewed again and we determined it was properly excluded from our study as it was a qualitative review and did not meet our inclusion criteria due to its study design.

b. We appreciate Reviewer #1 indicating that the Schumacher et al. (2007) reference was missing from the References list; this has been updated accordingly.

c. We appreciate the suggestion from Reviewer #1 to include the reference number for each citation in Table 2 and have updated accordingly.

d. We agree with Reviewer #1 that Table 2 could be re-organized to improve usefulness for the reader. As indicated above, we have now arranged Table 2 by cocaine use first, followed by amphetamine/methamphetamine use. We have additionally placed core reviews before supplementary reviews in the Table. This layout now also aligns with the organization of the Results section of the manuscript.

e. Please see comment above regarding how we have improved the layout of Table 2 and aligned it with the Results section of the manuscript. 

f. We appreciate the excellent observation from Reviewer #1 regarding the Discussion section, where we discussed some but not all evidence statements where we found insufficient evidence to support or discount the intervention. All of the interventions appraised as insufficient to support or discount their use are now included in the Discussion section, with recommendations for future research. Additional text reads as follows:

“CBT has been studied both alone and in combination with other psychotherapy interventions for treatment of stimulant use disorder, with some positive results. De Crescenzo et al. found a reduction in participant dropout with CBT alone, however when combined with CM, CBT was found to have a more pronounced effect, including in the outcome of participant abstinence (15). The available data for CBT in the treatment of stimulant disorder is limited, and more research is warranted to determine its clinical utility, focusing on a potentially more sustained effect when employed complementarily with contingency management psychosocial interventions, or pharmacologic treatment options.” (Discussion, page 34, 35)

“Similarly, for disulfiram, data supporting its clinical utility in treatment of stimulant use disorder is limited. The review by Pani et al. found positive results in reduction in use for one of four included RCTs (28). However, concerns regarding the safety of disulfiram, particularly in concurrent alcohol users, may limit its potential for future research (58).” (Discussion, page 35)

“Current data does not support the use of antidepressants for treatment of cocaine use disorder, although with regards to amphetamine use there is insufficient evidence to discount its use. Due to predominantly negative results for the use of antidepressants for treatment of cocaine use disorder, it may be possible to extrapolate from this data for the treatment of amphetamine use disorder (24, 25, 27). However, due to the unique mechanisms of action on presynaptic monoamine reuptake transporters, further research focused on the utility of antidepressants for amphetamine-type stimulants may be warranted.” (Discussion, page 35,36)

3. We appreciate the suggestions for reorganizing the text, especially the psychostimulant section. As suggested by the Reviewer and detailed above, the body of Results section in the manuscript is now organized by cocaine use first, followed by amphetamine or methamphetamine use. Studies classified as core reviews are also presented before supplementary reviews for each intervention type. 

4. We thank the Reviewer for pointing out that not all citations in Table 2 are found in the Appendix. The appendix (now entitled Appendix S3) has been updated with the remaining citations, and has additionally been reorganized to align with the Table 2 presentation of evidence statements. 

Minor Comments:

5. We appreciate the observation from the Review that line 94-96 did not fit well. We have revised this sentence to improve clarity:

“Of those accessing publicly funded treatment for substance use disorder in the United States, less than one in five individuals (17.8%) are doing so for cocaine or other stimulant treatment (8).” (Introduction, page 3)

6. We thank the Reviewer for identifying the use of DSM-V. Text now reads DSM-5. (Materials and methods page 5)

7. We agree with the Reviewer that the following line disrupted the flow of the text and it has been moved to the subsequent paragraph:

“Methods of searching and assessing primary literature, appropriateness of outcomes measured and conclusions, recognition of biases and steps taken to limit those biases, as well as the use of the PRISMA guidelines for systematic reviews were assessed.” (Materials and methods, page 8)

8. We thank the Reviewer for the suggestion to add the number of discrepancies that were resolved by a third reviewer. This has been added to the text as follows:

“There were two discrepancies which were resolved by a third independent reviewer (SN).” (Materials and methods, page 8)

9. We thank the Reviewer for identifying that Schumacher et al. was not included in the references. This has been updated. Schumacher et al. is a meta-analysis of four primary RCTs and met our inclusion criteria as supplementary evidence. 

10. We appreciate the observation by the Reviewer that the references were not aligned with the text. This has been updated accordingly. 

11. We agree with the suggestion from the Reviewer to change “were” to “was” (Results, page 18)

12. We appreciate the suggestion from the Reviewer to adjust the abbreviations used in the text. “SMD” was not spelled out at first use and has now been updated accordingly. “WMD” is spelled out at first use on page 21. “SMD” has been updated at first use as “standardized mean difference (SMD)” (Results, page 27)

13. We appreciate the Reviewer identifying the discrepancy in citation #55 and the citations have been updated accordingly. 

14. We thank the Reviewer for identifying that due to multiple reviews by Castells et al., it is unclear which review is being referred to. This has been updated in the text to include the date for clarity:

“Castells et al. (2016)” (Discussion, page 34) 

15. We thank the Reviewer for identifying inconsistencies in the formatting of the references. This has been updated accordingly. 

Reviewer #2: 

We appreciate the suggestion by the Reviewer to further discuss the definition of ‘recovery’ as it applies to standardized outcome measures. We have added to the Discussion section a description of changing views on how to define outcome measures in substance use research that goes beyond measures of abstinence (Discussion, page 36,37): 

“Another limitation of meta-analyses to date is the lack of standardized outcomes, making pooling of data difficult. Without evidence-based, standardized clinical trial outcomes for the treatment of stimulant use disorder, it will remain difficult to pool data and provide strong clinical recommendations. Long-term cessation of use has traditionally been the primary goal of substance use treatment and abstinence measures have been the most commonly implemented standard outcomes in randomized controlled trials (61). However, the definition of recovery may vary based on individual patient goals, by feasibility within the study time period, and may not always include abstinence (62). In 2015, the Analgesic, Anesthetic, and Addiction Clinical Trial Translations, Innovations, Opportunities, and Networks (ACTTION) group made recommendations for meaningful indicators of treatment success in future clinical trials on stimulant use, including a focus on the validation of patient reported outcome measures of functioning and the exploration of patterns of stimulant abstinence that may be associated with physical and/or psychosocial benefits (62). Future study may benefit from patient-oriented outcome measures, including psychosocial parameters such as improved social functioning, employment and acquisition of adequate housing.” (Discussion, page 36, 37)

We thank Reviewer #2 for the important suggestion to include details regarding subgroup effects of the interventions. We have added another table as supplementary material (Table S1) that provides more detailed information on population characteristics of each study, as well as any subgroup analyses that were reported. The following text was also added to the Discussion section: (Discussion, page 37, 38)

“It is important to note that the variability in treatment response among population subgroups is poorly addressed in the literature. Given the extensive variability in individual response to treatment, it is possible that an intervention appraised as ‘sufficient evidence to discount the effectiveness’ or ‘insufficient evidence to support or discount its effectiveness’ may have significant benefits for some patients or patient subgroups. In studies that performed subgroup analyses, including but not limited to age, sex, severity of substance use disorder and comorbid substance use disorders, meaningful results were limited. Future research would benefit from identifying those subgroups that may be more likely to benefit from certain interventions.” (Discussion, page 37)

“Subgroup analyses by Bhatt et al. demonstrated increased retention in treatments of longer duration (> 12 weeks) (37). Stratification by treatment duration moving forward may aid in identifying optimal and minimum effective treatment durations. Castells et al. (2016) found that psychostimulants increased abstinence and reduced cocaine use in those studies in which Attention Deficit Hyperactivity Disorder (ADHD) was not an inclusion criterion, which may be an important replicate moving forward (36).” (Discussion, page 37, 38)

We thank Reviewer #2 for raising the issue of people who use polysubstances. This is an important point to identify, as polysubstance use was the norm in the studies reviewed here. Table S1 has identified which reviews include participants with concurrent additional substance use disorders. We have added a section in the discussion regarding polysubstance use as follows: 

“Finally, polysubstance use is common (67, 68). Recent data demonstrates a rise in concurrent amphetamine and opioid use, and the role of stimulant use in the overdose epidemic remains poorly defined (69). Several of the included reviews identified that many participants suffered from polysubstance use disorder, however the effect of polysubstance use on treatment efficacy was rarely addressed. Future research would benefit from identifying the efficacy of interventions for stimulant use disorder specifically in the context of polysubstance use.” (Discussion, page 38)

General Comments:

Thank you for identifying the outdated World Drug Report citation. These statistics have been updated from the 2019 World Drug Report.

References have been corrected.

---

## [Decision Letter · Decision Letter 1]

3 Jun 2020

Treatment of stimulant use disorder: a systematic review of reviews

PONE-D-20-02936R1

Dear Dr. Fairbairn,

We’re pleased to inform you that your manuscript has been judged scientifically suitable for publication and will be formally accepted for publication once it meets all outstanding technical requirements.

Kind regards,

Kenji Hashimoto, PhD

Section Editor

PLOS ONE

Additional Editor Comments (optional):

Reviewers' comments:

Reviewer's Responses to Questions

**Comments to the Author**

1. If the authors have adequately addressed your comments raised in a previous round of review and you feel that this manuscript is now acceptable for publication, you may indicate that here to bypass the “Comments to the Author” section, enter your conflict of interest statement in the “Confidential to Editor” section, and submit your "Accept" recommendation.

Reviewer #1: All comments have been addressed

Reviewer #2: All comments have been addressed

2. Is the manuscript technically sound, and do the data support the conclusions?

Reviewer #1: Yes

Reviewer #2: Yes

3. Has the statistical analysis been performed appropriately and rigorously? 

Reviewer #1: Yes

Reviewer #2: N/A

4. Have the authors made all data underlying the findings in their manuscript fully available?

Reviewer #1: Yes

Reviewer #2: Yes

5. Is the manuscript presented in an intelligible fashion and written in standard English?

Reviewer #1: Yes

Reviewer #2: Yes

6. Review Comments to the Author

Reviewer #1: I reviewed the original submission of this manuscript and now have reviewed the resubmission. The authors have responded to all of my original comments. I thank them for their approach to documenting their responses in the response letter, as it greatly facilitated the re-review. I have reviewed the materials provided in the resubmission and have no additional major comments.

Minor comments

1. Line 143 – the original “DSM-IV” text was correct. (The switch from Roman numerals to Arabic or Western numerals has annoyed all of us!)

2. Line 243 – “Odd’s” does not require an apostrophe.

3. Lines 257-262 – there is an unformatted citation on these lines.

4. Table 2, bolded Psychostimulants row, line citing Perez-Mana et al 2013 – row is missing the numerical citation. I was unable to determine if the accompanying text (starting line 533) is accurate.

5. Please affirm Figure 1 and Appendix S2 Search Strategy was updated since the authors have now included De Crescenzo et al 2018.

Reviewer #2: The revised manuscript address the concerns noted in the original review. The manuscript organization and presentation of results has improved dramatically. The revised manuscript provides a valuable summary of what is known to date on the treatment of stimulant use disorder.

With that said, a few additional items need to be addressed:

FIGURE 1, the number of studies included is 28 in the figure but in the text it is 29. If it is 29, then the Full-text articles excluded would be 27. Please review Figure 1 in its entirety to ensure it matches the manuscript text.

INTRODUCTION

Line 82, “Stimulant use and stimulant use disorder is associated…”

• “is” should be “are”

Line 84-86 offers a global perspective that cocaine use has remained stable over the past decade however it may worth noting that many reports have identified an increase in cocaine use in the US during your study period. One example can be found here https://doi.org/10.1016/j.drugalcdep.2017.08.031

7. PLOS authors have the option to publish the peer review history of their article (what does this mean?). If published, this will include your full peer review and any attached files.

Reviewer #1: Yes: Robrina Walker

Reviewer #2: No

---

## [Editor Report · Acceptance letter]

10 Jun 2020

PONE-D-20-02936R1 

Treatment of stimulant use disorder: a systematic review of reviews 

Dear Dr. Fairbairn:

I'm pleased to inform you that your manuscript has been deemed suitable for publication in PLOS ONE. Congratulations! Your manuscript is now with our production department. 

Kind regards, 

on behalf of

Prof. Kenji Hashimoto 

Section Editor

PLOS ONE